# Public sector employment rigidity and macroeconomic fluctuation: A DSGE simulation for China

**Xiaodi Zhang**[ID]*

Institute of Economics, Shanghai Academy of Social Sciences, Shanghai, China

* zhangxd@sass.org.cn

## Abstract

Public sector employment in China has exhibited pronounced non-cyclical characteristics, with a recruiting scale and wage level showing limited responsiveness to economic fluctuations. The allure of civil service jobs in China has seen a significant resurgence post-COVID-19, with an observable increase in demand among educated job seekers for stable government positions amid growing economic uncertainties. This study investigates the implications of public sector employment rigidity on macroeconomic stability using a dynamic stochastic general equilibrium (DSGE) model integrated with search and matching (S&M) theory. Simulations incorporating alternative government job policies reveal that non-cyclical public employment exacerbates macroeconomic cyclical fluctuations. The low elasticity of public sector wages with respect to corporate wages fosters stable expectations among workers regarding the future value of government jobs, increasing the perceived value of the current state of unemployment. This leads job seekers to voluntarily remain unemployed, reducing labor supply to firms. Meantime, it preserves workers' bargaining power with firms, reinforcing wage stickiness and undermining the stabilizing role of price adjustments in employment. Hypothetical scenario analyses indicate that adopting a procyclical wage policy for the public sector can mitigate the obstacles of wage cuts for firms, stimulate the creation of new jobs during economic downturns, and consequently reduce the magnitude and duration of rising unemployment rates. In contrast, maintaining a non-cyclical public sector wage may not prevent a continuous rise in unemployment or a worsening economic situation, even with expanded sector recruitment. This finding holds significant relevance in the context of the post-COVID era characterized by an economic slump and employment tension, providing theoretical support for establishing a transparent and flexible wage adjustment mechanism in the public sector that is linked to market conditions.

## 1 Introduction

In the years following the COVID-19 pandemic, there has been a marked resurgence in enthusiasm for civil service careers in China [1]. Compared to tech companies, financial institutions, and foreign-invested enterprises, civil service has re-emerged as the dream job choice for well-

**Data Availability Statement:** The related code is held in the public repository of Figshare provided by Scientific Data (https://doi.org/10.6084/m9.figshare.26299456).

**Funding:** The paper received funding for this work provided by National Natural Science Foundation of China. The funders had no role in the study design, data collection, decision to publish, or manuscript preparation.

**Competing interests:** The authors have declared that no competing interests exist.

educated laborers [2]. The stability of working as a civil servant is highly valued amid soaring youth unemployment and massive private corporate downsizing [3]. A popular saying, "The summit of life is taking the civil service exam and getting in" indicates the top priority and desirability of government work in Chinese career plans [4]. A noticeable expansion in national civil service exam enrollment has occurred in the past five years from 2020–2024, with an annual average growth rate of 13.22%. In 2024, the exam placed about 40,000 individuals in jobs, which was roughly three times larger than that in 2019, the year before the COVID-19 pandemic outbreak in China [5]. The latest official statistics show a gigantic size of public employment at the end of 2022, including around 7 million formal civil servants in government units, 44 million in government-affiliated institutions, and about 40 million contracted workers temporarily employed by the two types of sectors. From a longer time interval perspective, public sector employment increased by as much as 700% between 1979 and 2023, with an annual growth rate above 4% and has generally been unaffected by the country's economic performance [6]. Stability, which is the greatest appeal of civil service jobs, means a modest but sustained paycheck that is guaranteed for a lifetime [7]; however, stability also means strict command and control over the size and compensation of public employment exerted by the central government [8]. In contrast to government layoffs and pay cuts in response to financial or debt crises in most countries, China's public labor market is completely non-cyclical [9], and not subject to the same market forces that confront private sector workers such as shocks, fluctuations, price signals, and expectations.

For a long time, Chinese job seekers have shown a strong preference for government employment, a tendency resulting from multiple factors [10, 11]. Historical and cultural consciousness has shaped and formed the psychological foundations of this preference [12]. China had established a "scholar-official"(士—官) civil service system since the Han Dynasty, and an imperial civil examination system(科举制) since the Sui and Tang dynasties, which institutionalized the "politics-education-politics" operational norm [13, 14]. This ingrained the ideal of "studying well to become an official"(学而优则仕) deeply into the collective consciousness of the Chinese people [15]. "Entering officialdom"(入仕) became the mainstream value pursuit for generations of people to achieve social mobility throughout Chinese history [16].

Influenced by Confucian culture, Chinese intellectuals' psychological traits include proactive spirit of "caring for the world",(心系天下), integrating the realization of personal values with participation in social affairs [17] and the pursuit of social well-being and the prosperity of the nation and ethnicity as well as the prosperity of the nation and ethnicity [18]. This unique recognition of government positions within traditional Chinese culture has endowed the role of a civil servant with high social prestige and status [19]. Under the influence of China's traditional cultural psychological structure, more educated individuals consider the added value of government positions as a significant symbol of achieving life aspirations [20]. Moreover, China's traditional "filial piety"(孝老) culture introduces a dismissive attitude toward elderly workers [21]. In contrast to the corporate sector, where workplace competition, age discrimination, and career bottlenecks contribute to the "35-year-old crisis" [22], these issues are not apparent or frequent in the government sector [23]. Instead, advantages may even exist for older employees [24].

From an individual economic perspective, pursuing a government job is a utility-maximizing choice [25]. Job seekers' rational behavior entails selecting lower-intensity, higher-benefit positions given sunk acquisition costs [26]. The civil service exam features low professional barriers, predictable question banks, and multiple choice of job options, incurring low trial-and-error costs and ample practice opportunities [27]. Once obtained, the job's overall utility exceeds the nominal salary [28]. Despite reforms, the "iron rice bowl" policy continues to ensure job stability for government employees [29]. According to the 2018 Civil Servant Law

of the People's Republic of China, civil servants can only be dismissed under five specific conditions [30]. The low dismissal and turnover rates in civil service provide strong job security, making government jobs synonymous with the "iron rice bowl" [31].

The "ironclad guarantees" system accompanies the "iron rice bowl" of civil servants [32]. Within China's political system, the government sector wields significant power and resource allocation capabilities, making government jobs typically offer stable income and superior welfare benefits [33]. For instance, the medical insurance reimbursement ratio and individual account contribution rate for civil servants exceed those of ordinary urban workers [34]. Additionally, civil servant subsidies can cover 10% of personal out-of-pocket expenses once medical costs reach a certain threshold, resulting in a stark disparity compared with the financial burdens of non-civil servant households during serious illnesses [29]. In China's housing provident fund system, government employers contribute at the maximum rate of 12%, which is a highly beneficial and stable arrangement [35]. In contrast, enterprise contribution rates fluctuate based on profit performance, impacting employees' major decisions on renting or buying a home [36]. Moreover, civil servants enjoy higher levels and efficiency in maternity insurance payments and occupational injury compensation than enterprise employees [37]. Therefore, China's current civil servants have shifted from the stereotype of being "stably poor" to closely aligning with the public's ideal of a desirable career path for a "drought and flood-proof" prestigious job [38].

From a social perspective, government employment exhibits significant path dependence at the family level [39]. Previous research indicates that "having a parent working in the public sector significantly increases the likelihood of the children securing their first job in the public sector after college graduation" [40]. Parents employed by the public sector can provide valuable information and network resources for their children's career success in civil service, representing a concrete manifestation of the inter-generational transmission of family cultural capital [41]. The family function theory considers civil service exams to be a gateway for college students to return to their hometowns and original family relationship networks in a stable manner, fulfilling the wishes of their elders and reconnecting previously fragmented living and emotional spaces [38, 42, 43]. Additionally, government jobs value human touch more than those in the private sector [44, 45], where lower turnover rates enable colleagues to understand one another more in-depth [46], thereby facilitating closer interpersonal networks [47]. This, in turn, provides individuals with more development opportunities [48] and social influence [49].

Since the COVID-19 pandemic, various objective factors such as slowing economic growth, sporadic outbreaks of geopolitical conflict, and changing international circumstances have led to a conservative attitude toward the job market for most college graduates [50–53]. In 2022, 77.35% of fresh college graduates reported considering taking the civil service exam, and 42.32% indicated that they would prefer to work for state-owned enterprises [54]. In 2023, the number of applicants for China's National Civil Service Exam reached a record high of over 3 million, with an average of more than 70 people competing for one position, and the most fiercely competitive position had a registration-to-admission ratio as high as 3572 to 1 [55]. For college graduates, "stably decent salary and benefits"(77.78%) and "high employment market pressure"(61.11%) are the main reasons for taking the civil service exam, and nearly 15% believe that family or friends also influence their decision [56]. As of 2023, the scale of China's civil service exam training market is about 33.1 billion yuan [57]. The preference for public sector jobs is even higher among youth group [58], with those born after 2000 showing a greater inclination toward public sector employment compared with those born in the 1990s [59]. During the pandemic, the government played a key role in crisis management [60], enhancing public awareness of the importance and stability of government jobs. This value

orientation has become more prevalent in post-pandemic career choices [61], enticing individuals' willingness to engage in government work that serves society [62]. Moreover, the pandemic has heightened people's emphasis on health and safety [63], and government jobs typically offer better working environments for physical and mental health [64].

Then the question is whether the stability of public employment is beneficial or detrimental to the stability of economic growth. Intuitively, maintaining a stable level of employment and wage in the public sector serves to cushion the impact of job loss in the private sector in an economic downturn, protecting employment and demand for investment and consumption from further contraction. However, this simple inference clearly ignores the labor market segmentation (LMS) caused by the iron rice bowl characteristics of the public sector and the consequent impact on households' decision making and private businesses' recruitment behavior. McDonald et al. (1985) underscores the importance of considering LMS when analyzing employment volatility and wage disparities [65]. Dickens et al. (1988) provides evidence suggesting that in a dual labor market a significant portion of well-educated laborers end up in the low-paid sector involuntarily, indicating a failure of market clearing [66]. By integrating search and matching frictions into standard New Keynesian model, the Diamond–Mortensen–Pissarides (DMP) model has emerged as a predominant framework for examining employment issues [67–69]. Hall et al. (2005) extend DMP by incorporating imperfect competition and wage stickiness [70]. The introduction of nominal wage rigidities offers a better explanation for fluctuations in inflation and output following monetary policy shocks [71, 72]. Additionally, wage stickiness has a substantial impact on hiring new workers [73], and the resulting productivity fluctuations in firms can in turn induce wage stickiness in equilibrium [74]. Ahn et al. (2023) notes that the division of labor equilibrium in the US leads to a three-tier segment that is marked by distinct employment stability and unemployment, and minimizes costs in response to economic turbulence [75]. A new global dataset examination demonstrates that public sector employment and compensation policies may have crowding-out effects on youth employment, particularly in low- and middle-income countries, where higher wage premiums are commonly observed in the public sector [76].

An article that is closely related to this study is Gomes (2014), who posits that the optimal public sector wage is contingent upon intersectoral frictions and should be responsive to business cycles to minimize unemployment volatility [77]. However, the unique LMS between the public and private sectors in China has distinct features in which the government has superior hiring standards, lower compensation, and limited positions, yet it has significant appeal for young, high-skilled laborers [78]. The Chinese central government initiated three civil service reforms since the 1980s aiming to enhance public servant performance by introducing a competitive selection process and merit-based pay [79]. However, deeply rooted cultural and ideological egalitarianism challenged establishing a significant performance-rewarding salary system [80]. The reforms enhanced public employee remuneration, marking a notable upward trajectory that moved away from the historical norm of modest compensation toward a more robust salary structure [81]. However, the wage growth still lagged behind other industries and the rising cost of living [82], and was commensurate with public employees' educational investment and skills [83, 84]. Nevertheless, the unique stability and security of civil service employment, along with government-promulgated values and goals, fosters numerous exam ronins, referring to young college graduates who are dedicated to repeated attempts at civil service examinations, withdrawing from the labor market [85]. Recent studies reveal the influence of Western theories and practices on China's civil service pay system, as decision makers strive for a balance between underpayment that may lead to corruption and overpayment that may erode public trust and regime legitimacy [86]. The rigidity of public employment is also inherent in limited promotional opportunities and mobility or transfer channels into or from other

sectors [87]. A future direction of reform includes a tenure system that allows regular dismissal and exit for government employees, aiming to enhance flexibility and mobility in civil service personnel management [88].

To evaluate the effects of job rigidity in the public sector on overall employment and economic fluctuation in China, this study employs the search and matching (S&M) model within the framework of The dynamic stochastic general equilibrium (DSGE) is a suitable choice To effectively capture the differences in employment behavior between the public and private sectors and predict their impact on economic cycles [89]. By integrating various mathematical and computational methods and algorithms, DSGE models can provide more accurate economic forecasts and policy analyses in diverse complex macroeconomic contexts [90], demonstrating strong applicability across fields and scenarios. The DSGE–vector autoregression (VAR) model that combines DSGE with VAR can be used for optimal public expenditure policy estimation, enhancing model robustness [91]. The DSGE-VAR model based on Bayesian Kalman Filter with Prior Update improves estimation accuracy for small samples and irregular data, with strength in analyzing systemic risks and cyclical fluctuations in specific industries [92]. Integrating Multilevel Sequential Monte Carlo (MLSMC) and Approximate Bayesian Computation (ABC) provides policymakers with a high-precision macroeconomic forecasting method under resource constraints [93]. The Next Reaction Method solves measurement error and omitted variable issues in dynamic macroeconomic models, increasing estimation accuracy and convergence speed [94]. The introduction of emerging technologies such deep learning and artificial neural network models (ANN) will also enhance the model's analytical and predictive capabilities in fields like public finance and financial market volatility [95, 96].

In this study, incorporating the search and matching (S&M) mechanism into dynamic general equilibrium system can yield various benefits. First, considering the career preference factor in workers' behaviors and choices aligns more closely with the realities of China. Workers choose between public and private sector jobs as preferred career paths, considering various market and institutional factors such as expected income, stability and security, sunk costs for searching, and related matters. Equilibrium in the labor market will be endogenously achieved based on maximization conditions and Nash bargaining over wage rates. Second, the general equilibrium analysis also makes it possible to conduct scenario simulation. Therefore, this study establishes a variety of hypothetical policy scenarios to examine the possible effects of introducing flexibility into the policy rules concerning the government's recruiting process.

## 2 Model setting

Controversies over the applicability of labor economic theory to China are commonly found in the literature [97–99], primarily due to the fact that the coexistence of multiple LMS types that violates the assumptions of complete competition and labor mobility [100–103]. Regarding to the focus of this study, the reality that diverges from theoretical assumptions is that Chinese job seekers have a strong preference for government-related jobs over private corporate positions, creating intangible barriers to labor mobility on one hand, and the rigorous competition in the civil service examination sets a high threshold for the public sector on the other hand, leaving a large number of job seekers unemployed. The S&M model enables a microscopic angle closer to socioeconomic realities in which the wage and unemployment rates are determined through the interdependence between workers searching for jobs with a certain preference and having bargaining power with private firms, and firms providing job openings conditional on government employment size.

This study adds China's unique employment market to the basic DSGE model by incorporating the following aspects. (i) The labor market is divided into parallel public and private

sectors. Workers search for jobs between the two sectors based on preferences, which are determined by the employment value function of the two types of jobs. (ii) The salary in the public sector is not confined to narrow book wage, but encompasses a wide range of tangible benefits that reflect the true remuneration and attractiveness of the position, which is represented in model calibration through wage premiums between sectors. However, due to their subjectivity, the intangible benefits of power, prestige, social status, and rent-seeking activities cannot be included in the analysis scope. (iii) Even if temporarily unemployed, job seekers that strongly prefer the public sector can choose to wait until they obtain satisfactory employment. Actually, recent years have seen a surging number of college graduates opting for full-time preparation for the civil service exam. (iv) Unemployment can have value for workers, including enjoying leisure, the emotional accomplishment of sticking to a predetermined goal, and hope for future success, which is represented by the unemployment value function, constituting workers' bargaining power in labor–capital negotiations.

## 2.1 Labor market

The model structure extends the research of Garibaldi et al. (2021) [104].The total labor force consists of three groups: unemployed $U$, public sector employment $L^G$, and private sector employment $L^P$. Unemployed workers find suitable positions to form new employment $N^i$, and frictional unemployment leads to a continuous fall in employment at the rate of $\mu^i$. Accordingly, the dynamic process of employment is $L_{t+1}^i = (1 - \mu^i)L_t^i + N_{t+1}^i, i \in \{P, G\}$, where $L_{it}$ denotes the current employment level.

   This study assumes that new employment is defined by the following Cobb–Douglas matching function:

$$N_t^i = A^i(U_t^i)^{\alpha^i}(V_t^i)^{1-\alpha^i}, i = P, G \tag{1}$$

where $U^i$ represents the unemployed individuals seeking employment opportunities in sector $i$. $V^i$ denotes the supply of vacancies in sector $i$. $\alpha^i$ represents the elasticity of job matching in sector $i$ with respect to the unemployed population. $A^i$ is defined as the efficiency of matching. The share of public sector job seekers is denoted by $\varphi_t = U_t^G/U_t$. $p_{1t}^i = N_t^i/V_t^i$ is the probability of a successful recruitment for sector $i$; $p_{2t}^i = N_t^i/U_t^i$ represents the conditional probability of successful employment for job seekers in sector $i$; and $p_{3t}^i = N_t^i/U_t$ is the unconditional job-finding rates.

## 2.2 Households

Households maximize utility by choosing private consumption $C_t$, public goods $G_t$ and unemployment status $U$. The utility functions of employment and unemployment are defined by in $f^E$ and $f^U$ respectively, the latter of which captures leisure and full-time preparation for the civil service exam. Households' decision making problem and the Euler equation obtained through first-order conditions are as follows:

$$max. \ E_t \sum_{t=0}^{\infty} \rho^t[f^E(C_t, G_t) + f^U(U_t)]$$

$$s.t. C_t + K_t = (1 + r_{t-1})K_{t-1} + \sum_i w_t^i L_t^i + T_t \stackrel{F.O.C.}{\rightarrow} \tag{2}$$

$$f_C^U(U_t) = \rho(1 + r_t)E_t[f_C^E(C_{t+1})]$$

where $E$ is the expected sign. $\rho$ is the discount factor. $r$ is the real interest rate. $K$ is households' capital holding. $w^i$ is the wage rate in sector $i$. $T$ represents the lump sum taxes used to finance the government's payroll.

## 2.3 Workers

The value function $W$ represents how individuals value their current status, which directly determines their job searching behavior.:

$$WE_t^G = w_t^G + E_t \rho_{t,t+1}\{[1 - \mu^G(1 - p_{2t}^G)]WE_{t+1}^G + \mu^G(1 - p_{2t}^G)WU_{t+1}\} \tag{3}$$

$$WE_t^P = w_t^P + E_t \rho_{t,t+1}\{[1 - \mu^P(1 - p_{2t}^P)]WE_{t+1}^P + \mu^P(1 - p_{2t}^P)WU_{t+1}\} \tag{4}$$

$$WU_t^i = f^U(U_t)/f^E(C_t) + E_t \rho_{t,t+1}[p_{2t}^i WE_{t+1}^i + (1 - p_{2t}^i)WU_{t+1}] \tag{5}$$

$$WU_t = max\{WU_t^G, WU_t^P\}$$

The value function of unemployment, Eq (5) is the key state variable influencing the workers' job seeking decisions [105], which is composed of two parts: (i) $f^U(U_t)/f^E(C_t)$, indicating that the higher the marginal utility of unemployment, the longer the job seeker can wait [106]. (ii) In period $t$, the unemployed have a probability of $p_{2t}^i$ to be reemployed and gain the value of $WE_{t+1}^i$ in period $t+1$. There is also a probability of $1 - p_{2t}^i$ to continue being unemployed with $WU_{t+1}^i$. The higher the future expected value of employment in sector $i$, the greater the value the worker places on unemployment status. A steady state of searching is reached when $WE^P = WE^G = WU$, yielding the endogenous proportion of public sector job seekers, $\varphi_t$ as follows:

$$[N_t^P E_t \rho_{t,t+1}(WE_{t+1}^P - WU_{t+1})]/[N_t^G E_t \rho_{t,t+1}(WE_{t+1}^G - WU_{t+1})] = (1 - \phi_t)/\phi_t \tag{6}$$

A rise in wage $w^G$ increases the value of public sector employment $WE^G$, attracting a larger proportion of job seekers targeting the public sector. As $\varphi_t$ increases, labor rushing into the public sector narrows the gap in marginal revenue between the two sectors, until working as a civil servant can no longer generate any additional return, and the search process reaches equilibrium.

## 2.4 Private firms

Assuming that a firm uses a single factor (labor) for production, and the output is proportional to the input of labor, then the production function is as follows:

$$Y_t = a_t^P L_t^P - \kappa^P V_t^P$$

where $\kappa^P$ represents the unit cost of providing vacant positions [107]. $y_t$ denotes the labor marginal product, $\partial Y_t/\partial L_t^P$. The value functions for a firm to provide job vacancies and to hire an employee are respectively as follows:

$$WV_t^P = \rho E_t[p_{1t}^P WJ_{t+1}^P + (1 - p_{1t}^P)WV_{t+1}^P] - \kappa^P \tag{7}$$

$$WJ_t^P = y_t - w_t^P + \rho E_t[(1 - \mu^P)WJ_{t+1}^P + \mu^P WV_{t+1}^P] \tag{8}$$

In Eq (7), the vacant position $V_t^P$ offered by a firm in period $t$ may be filled by qualified job seekers, generating a value of $WJ_{t+1}^P$ with the probability of $p_{1t}^P$; or it fails to secure any suitable candidates, in which case the vacancy $V_t^P$ becomes $V_{t+1}^P$ in the next period, generating a value of $WV_{t+1}^P$ with the probability of $1 - p_{1t}^P$. Hence, the expected total value of firms' recruiting is

$p_{1t}^P WJ_{t+1}^P + (1 - p_{1t}^P)WV_{t+1}^P$, minus its cost $\kappa^P$, yielding the net value $WV_t^P$. Eq (8) is similar in structure to Eq (7) where $WJ_t^P$ and $WV_t^P$ are mutually determined.

The state of equilibrium $WV_t^P = 0$ implies that providing job vacancies is no longer valuable for a firm and the firm ceases hiring activities. Combining Eqs (7) and (8) yields the following:

$$\kappa^P/p_{1t}^P = \rho E_t WJ_{t+1}^P, \quad WJ_t^P = y_t - w_t^P + (1 - \mu^P)\rho E_t WJ_{t+1}^P \qquad (9)$$

The equilibrium condition in Eq (9) indicates that the expected cost of providing job vacancies (on the left side of the equation) equals the expected return of hiring employees (on the right side). The expected return is subsequently influenced by marginal labor productivity $y$ and wage $w^P$.

The formation of wage $w^P$ follows the Nash bargaining process in which workers' $b$ is proportional to the unemployment value function $WU$ [108]. The decision making problem faced by the firm is as follows:

$$max_{w_t^P}(WE_t^P - WU_t)^b (WJ_t^P - WV_t^P)^{1-b} \xrightarrow{F.O.C.} \qquad (10)$$

$$bWJ_t^P = (1 - b)(WE_t^P - WU_t)$$

By integrating Eqs (4), (5), (9), and (10),] the firm's wage function is obtained as follows:

$$w_t^P = bY_t + [b\kappa^P p_{2t}^P (1 - \mu^P)]/p_{1t}^P \qquad (11)$$

## 2.5 Characterization of the steady state

The government utilizes linear labor technology to produce non-rivalry, free public goods $G_t = a_t^G L_t^G - \kappa^G V_t^G$. Similar to the private sector, the costs associated with posting job vacancies are deducted from production. The government collects tax $T_t$ and pays the labor wage $w_t^G L_t^G$. It also bears the recruitment cost $\kappa^G V_t^G$. The budget constraint balance condition for the government in period $t$ is as follows:

$$T_t = w_t^G L_t^G + \kappa^G V_t^G \qquad (12)$$

A decentralized equilibrium of the economy is obtained by combining all modules under the given policy rule of public employment, $\{V_t^G, w_{t+1}^G\}_{t=0}^\infty$, when the following conditions are met. (i) Households maximize the lifetime utility of unemployment $U_t$ (Eq (2)); (ii) Job seekers choose which sector to pursue, $\varphi_t$, to equalize the value of working in different sectors (Eq (6)); (iii) Firms choose the provision of job vacancies $V_t^P$ to ensure that the expected cost equals return (Eq (9)); (iv) Firms negotiate wages, $w_t^P$, with job candidates to satisfy the Nash bargaining criterion (Eq (10)); (v) The government sets the lump-sum taxes $T_t$ to accommodate the budget constraint (Eq (12)).

It can be proved that the government has the capacity to directly establish the optimal trajectory for job vacancies and to concurrently implement an appropriate wage policy, which induces the optimal proportion of public sector job seekers [109–111]. In a steady state, the private sector's vacancy level will be rendered optimal when workers' bargaining power is equivalent to the matching elasticity with respect to unemployment in the private sector ($b = \alpha^P$).

## 3 Calibration and steady-state variables

This study assumes a utility function of consumption taking the constant relative risk aversion (CRRA) form with intertemporal substitution elasticity equaling 1 and a linear form of the utility function of unemployment. According to the sectoral classification of the national economy developed by China National Bureau (GB/T 4754–2011), this study defines the public sector as public administration, social services and security, and social organizations, which includes six major types of jobs: (i) The Chinese Communist Party units; (ii) Government authority and administrative bodies, courts, and procuratorates; (iii) The Chinese People's Political Consultative Conference and democratic parties; (iv) Social services and security departments; (v) Citizens' organizations and social organizations; and (vi) Urban and rural community autonomous organizations. The operation and personnel costs of these institutions are financially supported by by public finance and managed following the Civil Servants Law of China. In this study, the term "nonpublic sector" describes all private, collective, and state-owned enterprises.

A brief explanation of the rationale for the steady state values of the main variables in the model is provided. Although it is a more common practice to use quarterly or monthly data in DSGE models, since the vast majority of data required in this study are only available at an annual level, this study calibrates parameters based on China's annual data from 1985 to 2022. (i)The employment size of the public sector, $L^G$, is obtained from the Number of Employees at the End of Each Fiscal Year listed in the China Labor Statistical Yearbook. (ii) Public sector employees' average yearly remuneration, $w^G$, is measured by dividing the total amount of public administrative expenditure minus foreign affairs expenditure (before 2007) and general public service expenditure minus interest payment on domestic and foreign debts (after 2007) by $L^G$ to cover various forms of monetary benefits received by civil servants, including basic salaries, performance bonuses, housing/medical/transportation/meals allowances, monthly/quarterly/annual bonuses, travel subsidies, overtime pay, and retirement pension. (iii) Private sector employment size $L^P$ is obtained by summing the employees from state-owned enterprises, collective enterprises, private enterprises and household-run businesses listed in the China Labor Statistical Yearbook. (iv) The average per capita wage in private sector, $\bar{w}^P$, is obtained through weighting the wage indices of private employees by sector from the China Statistics Yearbook. (v) Steady state values are derived from sample averages. The estimated public sector wage premium compared with all private paid employees, which is denoted as $\bar{w}^G/\bar{w}^P$, is approximately 1.5849. The parameters for technology shocks are calibrated based on the methodologies outlined in Christiano et al. (2016) [112] and Lu et al. (2024) [113], with the autoregressive coefficient and standard deviation set at 0.95 and 0.0071, respectively. Table 1 presents the values used to calibrate the model.

## 4 Benchmark simulation: Economic fluctuation under public employment rigidity

### 4.1 Measuring public employment rigidity

This study divides the 1985–2022 sample period into three subintervals based on three significant policy reforms in civil service pay introduced by the State Council of China. The 1985 reform set wage levels that were strictly controlled by the central government. Since this reform, the wages in publicly funded organizations have remained below the national average for a long time. In the 1993 reform, four policy initiatives specifically aimed at increasing public sector employment. These included establishing a structural wage system with pay variations between positions, levels, ranks and regions; enabling stepwise salary growth by rank, level, and length of service; implementing a special allowance for underdeveloped areas; and

**Table 1. Calibrated parameters of the model.**

| Description | Parameter | Source |
|---|---|---|
| !Job separation rate | $\mu^i$ | Approximated by the inverse of job tenure. An analysis of the China Health and Nutrition Survey (CHNS) data reveals that the average job tenure for government and enterprise positions are 11.3 years and 3.8 years, respectively [114]; therefore, job separation rates are assigned to be 8.85% for the public sector and 26.32% for the enterprise sector. This indicates that the separation rate in the private sector is three times that of the public sector, which is consistent with the *Turnover and Salary Adjustment Research Report* by 51job Research Institute (China's largest job matching platform) [115]. |
| Proportion of laborers seeking jobs in the public sector | $\varphi_t$ | Approximated using the average ratio of annual national civil service examination registrants to the total number of university graduates in the sample [5]. |
| Natural level of unemployment rate | $\bar{U}$ | Assigned 0.052 in alignment with the average urban surveyed unemployment rate in China [116]. |
| Matching elasticity with respect to unemployment | $\alpha^i$ | Derived from a regression model with the sectoral job-finding rate's logarithm as the independent variable and the tightness ratio's logarithm (job openings to unemployment) as the dependent variable. The elasticity is 0.5 for the private sector (in line with estimates from the literature [117, 118]) and is 0.128 for the public sector in line with the average civil service exam admission rate in China [5]. |
| Probabilities of successful recruitment and getting a job | $p_{1t}^i$ $p_{2t}^i$ $p_{3t}^i$ | Firms' job filling rate is set to 0.312 and workers' job securing probability in the private sector is set to 0.703 to align the active job openings-to-applicants ratio with an average of 1.028, consistent with the mean observed in empirical data in 1985–2020 from the WIND database [119]. The government's job filling rate is set to 0.743 and workers' job securing probability in the public sector is set to 0.036 to align the average admission rate and vacancies-to-applicants ratio in China's civil service exam [120]. The unconditional job finding rate is set to 0.035 for the government sector and 0.523 for firms to align the new hiring size for each sector [116]. |
| Matching efficiency | $A^i$ | Calibrated to reproduce $p_1^i$ ($A^G = 0.7432, A^P = 0.3126$) |
| Unit cost of posting job vacancies | $\kappa^i$ | Set to 0.01 in the public sector and 0.014 in the private sector. The costs of recruiting a civil servant is approximated by the per capita expenses to organize civil service exams published by the Ministry of Human Resources and Social Security of China [121]. Then firms' average cost is 40% higher than that of the government, which is consistent with the Chartered Institute of Personnel Development survey [122], International Labor Organization [123], National Audit Office [124]. |
| Nash bargaining power | $B$ | Set to 0.4 to satisfy the Hosios condition [125] |
| Discount factor | $\rho$ | Set to 0.9855, implying an annual interest rate of 5%, and Technology in both sectors is normalized to 1. |

introducing an annual lump-sum performance bonus. The 2006 reform took three key actions, which included introducing a regular cost-of-living allowance in addition to the base wage and year-end bonus, setting a nationally unified standard for rank and level salaries, and subsidizing low-level and grassroots civil servants. The 1993 reform may have enhanced the attractiveness of public jobs by breaking the low-level egalitarianism since 1985. The 2006 reform took further steps to reduce the gaps between positions and regions, increasing laborers' motivation to pursue public sector employment.

Table 2 presents a preliminary examination of the correlations between public sector employment ($L^G$), non-public sector employment ($L^P$) and GDP using diverse measures that

**Table 2. Correlations between public and private employment sizes and GDP.**

| | 1985–1993 | 1994–2006 | 2007–2022 | 1985–2022 |
|---|---|---|---|---|
| $Corr(L^P, GDP)$ | 0.8519 | 0.8613 | 0.8881 | 0.8740 |
| $Elas(L^P, GDP)$ | 0.8440 | 0.8135 | 0.8527 | 0.8476 |
| $Std(L^P)/Std(GDP)$ | 0.8888 | 0.8984 | 0.9318 | 0.9053 |
| $Corr(L^G, GDP)$ | 0.1428 | 0.1617 | 0.1544 | 0.1518 |
| $Elas(L^G, GDP)$ | 0.1273 | 0.1497 | 0.1451 | 0.1298 |
| $Std(L^G)/Std(GDP)$ | 0.2628 | 0.3312 | 0.4280 | 0.3126 |
| $Corr(L^G, L^P)$ | 0.1887 | 0.2124 | 0.2270 | 0.2191 |
| $Elas(L^G, L^P)$ | 0.2207 | 0.2467 | 0.2527 | 0.2441 |
| $Std(L^G, L^P)$ | 0.4832 | 0.5641 | 0.6655 | 0.5753 |

**Table 3. Correlations between public and private sector wages and GDP.**

|  | 1985–1993 | 1994–2006 | 2007–2022 | 1985–2022 |
|---|---|---|---|---|
| $Corr(w^P, GDP)$ | 0.5102 | 0.5559 | 0.6049 | 0.5496 |
| $Elas(w^P, GDP)$ | 0.2714 | 0.3678 | 0.4792 | 0.3357 |
| $Std(w^P)/Std(GDP)$ | 0.5346 | 0.8213 | 0.8352 | 0.7026 |
| $Corr(w^G, GDP)$ | 0.1649 | 0.3777 | 0.2900 | 0.1757 |
| $Elas(w^G, GDP)$ | 0.1415 | 0.3417 | 0.2056 | 0.1852 |
| $Std(w^G)/Std(GDP)$ | 0.2809 | 0.3289 | 0.5050 | 0.3509 |
| $Corr(w^G, w^P)$ | 0.1153 | 0.2757 | 0.1920 | 0.1347 |
| $Elas(w^G, w^P)$ | 0.1344 | 0.2495 | 0.1906 | 0.1891 |
| $Std(w^G)/Std(w^P)$ | 0.3754 | 0.6104 | 0.5881 | 0.4933 |

include correlation coefficient, elasticity, and standard deviation ratio. The elasticity coefficient is obtained by estimating $\beta_1$ in $\log y_t = \beta_0 + \beta_1 \log x_t + \zeta_t$. The high values (around 0.8–0.9) of all indicators in each sub-interval imply a high degree of procyclicality between $L^P$ and GDP. Conversely, $L^G$ shows minimal correlation to GDP despite certain interdependence with $L^P$.

Table 3 compares the correlation of public sector wage $w^G$ and non-public sector wage $w^P$ with GDP. In contrast to the non-cyclical characteristics of $L^G$ in all sub-intervals, the correlation between $w^G$ and economic growth in 1994–2006 was more prominent than in other periods. Possible explanations are twofold. (i) Before 2006, local governments had some autonomy to adjust officials' bonuses and allowance based on fiscal revenue [126], which was highly correlated with GDP. In addition, the central government improved civil servants' wage standard four times from 1997 to 2003 [127], strengthening the synchronization between public sector salaries and national economic growth; however, the wage standard has not changed since 2006 [128]. (ii) The Asian Financial Crisis in 1998 prompted the Chinese government to embark on an expansionary fiscal policy to stimulate domestic demand and combat deflation [129]. The local government subsequently raised expenditure, driving up the income level of civil servants.

## 4.2 Simulated employment and output volatility under rigid public employment policy

This section introduces the rigidity of public sector employment into DSGE models to simulate the fluctuation (standard deviation) of macroeconomic variables such as overall employment and output. The simulated values are then compared with a hypothetical baseline scenario excluding the public sector. If the variables display higher volatility in the reality scenario than in the baseline scenario, the presence of public sector employment reducing economic stability is corroborated.

The cyclicality of public sector employment to growth can be measured by its elasticity to private sector employment, which is formulated as follows [130, 131]:

$$ln(V_{t+1}^G) = ln(\bar{V}^G) + \chi^V[ln(V_t^P) - ln(\bar{V}^P)] \tag{13}$$

$$ln(w_{t+1}^G) = ln(\bar{w}^G) + \chi^u[ln(w_t^P) - ln(\bar{w}^P)] \tag{14}$$

where $\bar{V}, \bar{w}$ represent the steady state values of employment and wage levels, respectively. The above two equations are used to depict the degree to which the public sector responds to changes in market conditions. The sign and value of the elasticity parameters $\chi^V$ and $\chi^w$

**Table 4. Volatility of employment and output simulations under various policy regimes.**

| | Including the public sector | | | | Excluding the public sector |
|---|---|---|---|---|---|
| | 1985–1993 policy | 1994–2006 policy | 2007–2022 policy | 1985–2022 policy | |
| **Standard deviation** | | | | | |
| Employment | 1.3844 | 0.8002 | 0.9148 | 1.0863 | 0.2973 |
| Total output | 2.2842 | 1.6920 | 2.1488 | 1.8391 | 1.2215 |
| Average wage | 0.4308 | 0.8628 | 0.6903 | 0.7636 | 0.9872 |
| Private employment | 1.6062 | 0.9645 | 1.0425 | 1.1585 | 0.2973 |
| Public employment | 0.5563 | 0.2619 | 0.3171 | 0.3315 | N/A |
| Private wage | 0.8900 | 0.8845 | 0.9105 | 0.8784 | 0.9872 |
| Public wage | 0.3536 | 1.0880 | 0.5708 | 0.4946 | N/A |
| **Public–private sector elasticity** | | | | | |
| Wage | 0.1343 | 0.2493 | 0.2004 | 0.1890 | NA |
| Employment | 0.2385 | 0.2385 | 0.2385 | 0.2385 | NA |

correspond to different policy rules. $\chi > 0$, $\chi < 0$, $\chi = 0$ respectively represent the public employment adjusting pro-cyclically, counter-cyclically, and being unaffected, where a larger the value of $|\chi|$ indicates higher sensitivity.

A reality scenario can be constructed using historical data on public and private sectors' employment elasticity (as shown in Tables 1 and 2). As shown in the previous section, the rigidity of public employment size remains similar across different subintervals; therefore, the value of $\chi^v$ is taken as an average of 0.2441. However, the correlations between public sector wage and GDP across subintervals exhibit significant differences; thus, $\chi^w$ takes values of 0.1344, 0.2495, 0.1906, and 0.1891 for 1985–1993, 1994–2006, 2007–2022, and 1985–2022, respectively. The hypothetical baseline scenario excluding the public sector is described by $\bar{N}^G = 0$.

The random shock on firms' production follows an AR(1) process with autoregressive coefficient $\psi = 0.95$, that is, $ln\,\eta_t^P = (1 - \psi)ln\,\bar{\eta}^P + \psi ln\,\bar{\eta}_{t-1}^P + \varepsilon_t^\eta$. This study takes logarithmic linearization of the variables and removes trends using the Hodrick–Prescott (H–P) filter to yield short-term fluctuations. To measure the volatility of each variable, a one-order autoregression is then applied to obtain the standard deviation of the fluctuations (Table 4), revealing two main conclusions. (i) As the elasticity of public sector wages decreases, the volatility of firm employment and total employment increases, indicating that greater public sector rigidity diminishes labor market and economic stability. (ii)When $\chi^w$ is taken as the average value of the whole sample range of for 1985–2022, the standard deviations of employment and output are 1.0863 and 1.8391, respectively. Compared with the hypothetical baseline scenario excluding the public sector, employment volatility increases by 75%, and that of output increases by 50.56%. In other words, the public sector actually reduces the stability of the labor market and total output.

To assess the model's explanatory power, Table 5 calculates the cross-correlation coefficients and relative standard deviations (based on simulation results) for each variable and total output (Simulation column) and compares them with the results calculated based on historical data (Real data column, which corresponds to $Corr(w^i, GDP)$, $Corr(L^i, GDP)$, $Std(w^i)/Std(GDP)$, $Std(L^i)/Std(GDP)$, and $i = P, G$, as presented in Tables 1 and 2). The results demonstrate that the deviation rate is no more than 50%, indicating that the simulation of employment scale and wage fluctuations aligns well with economic reality in both direction and amplitude.

In contrast to intuitive expectations, the simulation indicates that employment in China's public sector, which is detached from economic performance and market forces, does not actually act as a stabilizer for the national economy, but rather exacerbates the deviation of

Table 5. Model it: Real data versus benchmark model simulation.

| | Corr (X, GDP) | | | Std (X)/Std(GDP) | | |
|---|---|---|---|---|---|---|
| | Simulation | Real data | Simulation/real | Simulation | Real data | Simulation/real |
| Employment | 0.5411 | 0.6598 | 0.8314 | 0.5990 | 0.8680 | 0.7010 |
| Public employment | 0.1128 | 0.1483 | 0.7701 | 0.1828 | 0.3055 | 0.6068 |
| Private employment | 0.8371 | 0.8542 | 0.9939 | 0.6387 | 0.8848 | 0.7320 |
| Average wage | 0.3784 | 0.4252 | 0.9046 | 0.4210 | 0.6422 | 0.6648 |
| Private wage | 0.5108 | 0.5493 | 0.9413 | 0.4843 | 0.7021 | 0.6994 |
| Public wage | 0.1493 | 0.1756 | 0.8668 | 0.2727 | 0.3506 | 0.7887 |

growth from the steady state. In this regard, potential mechanisms can be examined based on the structure of the model.

When the economy is subject to a negative production shock, firms pursuing profit maximization will cut job supply. Without a public sector, the difficulty of finding a job rises, and the prolonged duration of unemployment decreases in the unemployment value function. Concerning job hunters' bargaining, a lower unemployment value weakens workers' power, enabling firms to exercise greater authority in wage negotiation [132]. Therefore, workers must accept reduced wages to preserve employment. Reduced wages decrease firms' cost of creating jobs, strengthen their willingness to expand hiring, and prevent the unemployment rate from persistently rising higher than the steady state value.

However, a public sector that exhibits strong non-cyclical characteristics supports workers' expectations that even if the economy deteriorates, public sector remuneration and recruitment will not decline. Consequently, the attractiveness of public sector jobs increases and that of the private sector decreases. This shift would result in more voluntarily unemployed individuals and facilitate an exodus from the private sector in pursuit of government positions. As the amount of public sector recruitment is strictly controlled by the central government, it remains stable, resulting in a rising number of unemployed job seekers. The temptation of sustained pay in the public sector causes job seekers to accept temporary unemployed status in pursuit of better occupations in the future. In summary, public sector employment rigidity stabilizes the expected value of unemployment, enhancing job seekers' willingness to wait for preferred public sector employment. Once the determination of unemployed workers to maintain unemployed status increases, they will not easily accept low contract wages offered by firms, increasing their bargaining power, which will subsequently challenge firms' intention to expand production at a lower cost.

At this juncture, private businesses are disrupted by elevated labor costs and a contraction in labor supply. Under these dual pressures, firms become more hesitant to create jobs, deviating short-term employment levels from the long-term natural rate. In summary, the stability of public sector employment weakens the labor market's self-correction mechanism in response to shocks, which aggravates employment fluctuation and output at the macrolevel, where more rigidity imposes higher volatility (as shown in Table 3, the maximum volatility of employment and output occurred in 1985–1993).

## 5 Hypothetical scenario simulation: Economic fluctuation under flexible public employment policies

This section constructs five distinct scenarios that characterize the potential responses of the public sector to macroeconomic fluctuations by altering the values of elasticity parameters $\chi^V$ and $\chi^w$, and still imposes a 1% negative technical shock following an AR(1) process.

Scenario 1: Public sector wage and employment size are assumed to be non-responsive to cyclical fluctuations ($\chi^w = 0$, $\chi^V = 0$), implying that the government exercises complete control over the supply–demand dynamics of civil servant positions, thereby decoupling public sector employment from the business cycle.

Scenario 2: Public sector wage is non-cyclical, while the employment size exhibits a counter-cyclical pattern ($\chi^w = 0$, $\chi^V = -1$), suggesting that the government endeavors to sustain employment during economic downturns, as a potential counterbalancing measure to mitigate the effects of unemployment.

Scenario 3: Public sector wage is positively correlated with economic fluctuations, and employment size remains non-cyclical ($\chi^w = 1$, $\chi^V = 0$), reflecting cost containment measures that align with fiscal constraints during economic stress.

Scenario 4: Public sector wage and employment size are pro-cyclically adjusted ($\chi^w = 1$, $\chi^V = 1$), suggesting that the government resorts to layoffs and wage reductions as a strategy to cope with economic and fiscal pressures, aligning labor market adjustments with the prevailing economic conditions.

Scenario 5: A pro-cyclical wage adjustment is coupled with a counter-cyclical hiring policy ($\chi^w = 1$, $\chi^V = -1$), addressing fiscal challenges by cutting wages while simultaneously expanding job vacancies to preserve overall employment stability despite economic downturns.

## 5.1 Volatilities under different policy scenarios

Using the H–P filter for detrending, we obtain the standard deviations of short-term fluctuations in employment and output for each scenario (Table 6), which reflects the magnitude and duration of deviations from the long-term steady state. A higher standard deviation indicates more pronounced and sustained volatility in response to stochastic shocks. The simulation results reveal that the volatility of employment and output in Scenarios 1 and 2 is significantly higher than that of Scenarios 3–5, and exceeds that of the reality policy scenario presented in Table 3. This implies that economic fluctuations may intensify when the public sector fails to implement effective countermeasures. In contrast, in Scenarios 3–5, the stability of employment and output is significantly enhanced through flexible adjustments in employment policies, particularly wage policies, approaching or even surpassing the benchmark scenario without public sector intervention. This finding underscores the pivotal influence of public employment on macroeconomic management, demonstrating that the government can act as an effective "automatic stabilizer" against shocks by implementing appropriate wage and hiring policies.

Public sector employment elasticity is a determinant of workers' expectations regarding the future value of employment and their assessment of the current value of unemployment,

Table 6. Simulation results under different policy scenarios.

| | Scenario 1 | Scenario 2 | Scenario 3 | Scenario 4 | Scenario 5 |
|---|---|---|---|---|---|
| **Employment** | $\chi^w = 0$, $\chi^V = 0$ | $\chi^w = 0$, $\chi^V = -1$ | $\chi^w = 1$, $\chi^V = 0$ | $\chi^w = 1$, $\chi^V = 1$ | $\chi^w = 1$, $\chi^V = -1$ |
| **Total output** | 1.4128 | 1.1714 | 0.3812 | 0.4428 | 0.2540 |
| **Average wage** | 2.4267 | 1.7088 | 1.2320 | 1.3058 | 0.9831 |
| **Private employment** | 0.1760 | 0.0953 | 0.7518 | 0.9557 | 0.6907 |
| **Public employment** | 1.8787 | 1.6764 | 0.4666 | 0.4428 | 0.4907 |
| **Private wage** | 0.0000 | 1.6764 | 0.0000 | 0.4428 | 0.4907 |
| **Public wage** | 0.2686 | 0.1807 | 0.7518 | 0.9557 | 0.6907 |

which ultimately influences job seeking decisions. A detailed examination of each scenario based on the theoretical model in section 2 is as follow:

In Scenario 1, the rigidity of the public sector wage $w^G$ has two notable impacts. (i) Given that the public sector wage $w_{t+1}{}^G$ will not decrease in period $t$+1, the future public sector employment value function $WE_{t+1}^G$ also remains at a high level according to Eq (3). Eq (5) indicates that a higher the expected value of employment in $t$+1 will produce a higher current unemployment value function $WU^G = WU$. In other words, a rigid $w^G$ causes workers to choose unemployment even in an economic downturn. Eq (10) indicates that the stability of $WU$ prevents a sharp decline in workers' wage bargaining power b as market conditions deteriorate. Current employees can use resignation as a threat to deter enterprises from cutting salaries [133], hindering firms from absorbing more workers and increasing job seeking difficulties. (ii) Based on Eqs (4) and (6), the relative decline in firms' compensation $w^P/w^G$ decreases the proportion $1-\varphi_t$ of job seekers pursuing employment in the private sector. Consequently, the unemployed feel empowered to reject low-income jobs, which diminishes firms' enthusiasm for job creation.

In Scenario 2, the public sector proactively increases its recruitment scale to mitigate unemployment, but this approach unfortunately fails. Expanding government position offerings $V^G$ elevates the likelihood of obtaining public sector employment, which is represented by $P_2^G$. As per Eq (3), the uptick in the public sector employment value function $WE^G$ draws a larger number of unemployed and firm workers toward public sector positions, resulting in a larger preference ratio $V^G$ than in Scenario 1. The unemployment value function $WU$ remains stable or even improves, which bolsters workers' bargaining power b. While the private sector's capacity to hire shrinks, the public sector accommodates a subset of job seekers through recruitment expansion. Therefore, the standard deviation of employment and output fluctuations in Scenario 2 is lower than in Scenario 1.

In Scenario 3, $WU^G$ and $WU^P$ decline simultaneously, and the public sector no longer stabilizes the value of unemployment, which decreases workers' bargaining power. This shift enhances firms' dominance in wage negotiations, reduces workers' resistance to wage cuts, and increases firm's incentive to create new employment opportunities. Table 5 reveals that Scenario 3 exhibits a 73.02% and 67.46% reduction in overall employment volatility compared with Scenarios 1 and 2, respectively. Scenario 4 introducing pro-cyclical adjustment of employment size does not exhibit significant differences compared with Scenario 3. Scenario 5, which implements counter-cyclical adjustments, achieves the lowest output and employment volatility among all scenarios by facilitating intersectoral labor force mobility.

## 5.2 Responsive paths of employment variables to shocks

Fig 1 depicts the impulse responses of key endogenous variables to a 1% negative productivity shock under each scenario through the path of "firm wage → firm job offering → workers' job preference → firm employment → overall employment" where the elasticity of public sector employment has a pivotal role [134].

The dynamic response trajectory of firm wages $w_t^P$ is illustrated in Fig 1(A). When the technology shock occurs ($t = 0$), due to the constraints of labor contracts, wages cannot immediately adjust, exhibiting a lag. Subsequently, workers' differential bargaining power ($b$) leads to divergent wage paths across scenarios. In Scenarios 1 and 2, public sector wages remain unaffected by the technological shock, maintaining the stability of the unemployment value function ($WU_t = WU_t^G$), with Scenario 2 even exhibiting an increase in the unemployment value function alongside the rise in public sector employment probability ($p_2^G$). Hindered by workers' bargaining power ($b$), firms' wage reduction is slow and the extent of pay cuts is limited,

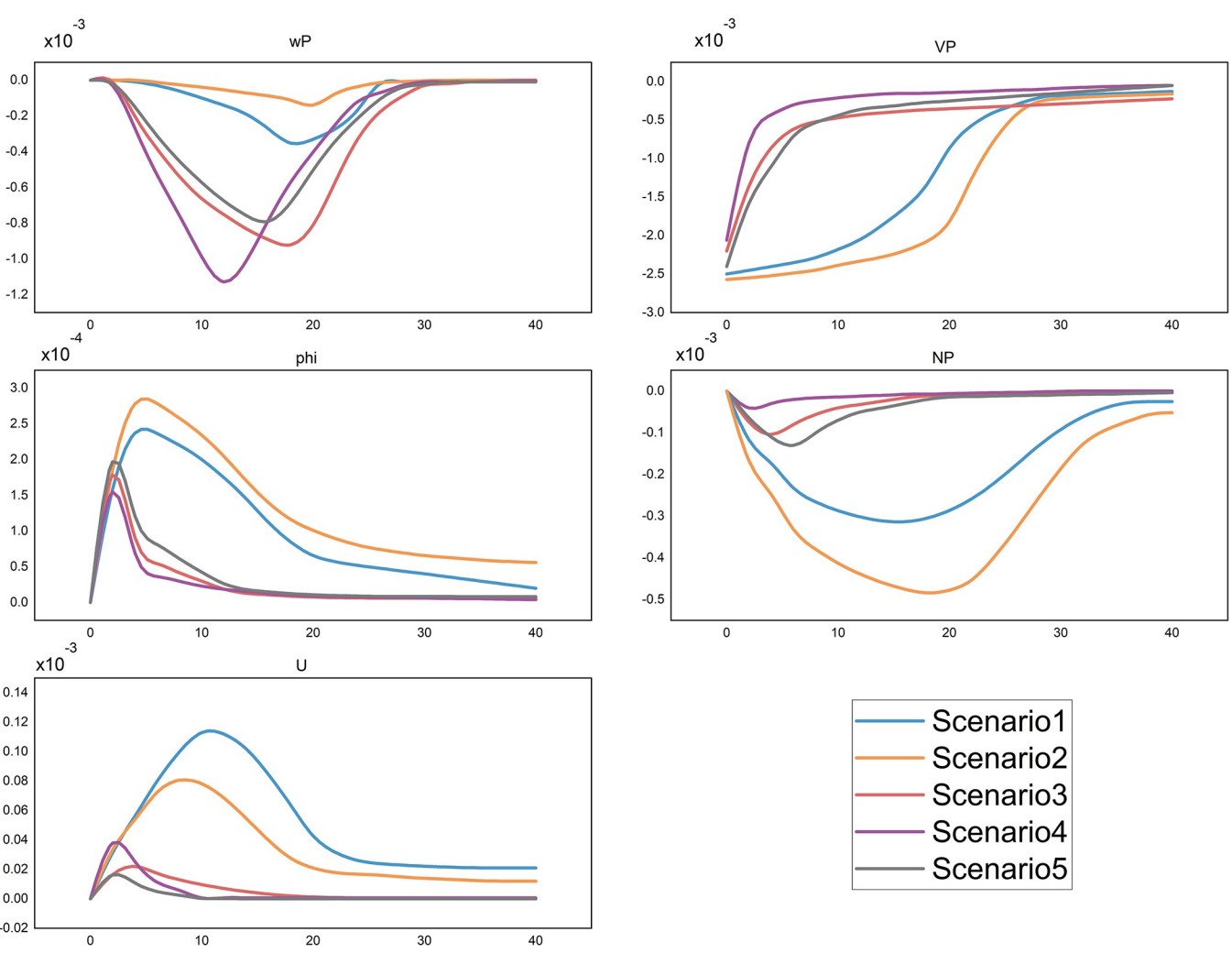

**Fig 1. Effects of technology shock on employment variables.**

which is supported by Eq (11). However, $b$ is not permanently stable [135–137] because the utility of unemployment ($f_t^U$) diminishes as unemployment duration extends, and the increase in public sector job seekers ($U^G$) drives down the employment probability ($p_2^G$), resulting in a fall in $WU_t^G$ and producing a faster downward adjustment of wages in Scenarios 1 and 2 after five periods. In contrast, in Scenarios 3–5, public and private sectors reduce wages proportionally and in the same direction, simultaneously lowering the unemployment value functions ($WU_t^G$ and $WU_t^P$), which diminishes workers' bargaining power in wage negotiations and enables firms to easily compress wage expenditure. Scenario 4, with a procyclical public employment policy, has the lowest postshock public sector employment probability ($p_2^G$), prompting workers to value private sector positions more, which results in the fastest wage decline. By increasing the public employment probability ($p_2^G$), Scenario 5 retains some bargaining power for workers, resulting in the slowest wage decline, and Scenario 3 is between Scenarios 4 and 5. According to Eq (9), the rebound of $y_t^P - w_t^P$ indicates an increase in the expected return of hiring new employees ($E_t WJ_{t+1}$), and a resurgence in the motivation to create new jobs ($V_t^P$). The probability of successful employment in firms ($p_2^P$), which increases alongside $V_t^P$, will elevate workers' unemployment value function ($WU_t = WU_t^P$) and

bargaining power ($b$), subsequently causing a recovery in the wage rate ($w_t^P$). Consequently, wages rebound in all scenarios after falling to a certain level.

Fig 1(B) presents the response of firms' job vacancy supply ($V_t^P$), which is directly contingent upon the wage costs ($w_t^P$). In contrast to the wage stickiness associated with labor contracts, the provision of new positions is entirely at firms' discretion. At the time of the technology shock, according to Eq (9), a decrease in marginal productivity ($y_t^p$) reduces a reduction in the marginal return on employment ($WJ_t^P$), prompting an immediate decline in job vacancies ($V_t^P$) at t = 0 in all scenarios. In Scenarios 1 and 2, the slow reduction in $w_t^i$ diminishes firms' motivation to expand employment. Particularly in Scenario 2, with increased attractiveness of public sector employment, firms find it more difficult to recruit ($p_1^p$ falls), reducing the value of job offering ($WV_t^P$) as shown in Eq (8), and further decreasing the value of employment ($WJ_t^P$), resulting in sluggish recovery in job vacancies. In contrast, in Scenarios 3–5, the rapid reduction in labor wage costs increases firms' willingness to create jobs and drives a swift rebound in vacancy supply within shorter periods.

Fig 1(C) delineates the responses in the proportions of job seekers pursuing jobs in the public sector ($\varphi_t$). Labor mobility is primarily guided by the relative wages and the quantity of job vacancies offered in the two sectors. From t = 1 onward, in Scenarios 1 and 2, a reduced firm job supply ($V_t^P$) and an increased relative public sector wage ($w^G/w^P$) causes a rise a significant rise in $\varphi_t$. In Scenarios 3–5, where wages in both sectors are fully flexible, the increment is lower than that observed in Scenarios 1 or 2. Then, $\varphi_t$ continues to decline in Scenarios 1 and 2, while Scenarios 3–5 return to preshock levels at a faster pace.

Fig 1(D) presents a comparison of new matching of firm employment ($N_t^P$), which is jointly determined by firm vacancy supply and the pool of job seekers targeting firms ($N_t^i = A^i (U_t^i)^\alpha (V_t^i)^{1-\alpha}$). In Scenarios 1 and 2, persistent high wages and job seekers' high preference for public sector employment contracts labor supply and demand, causing an inevitable decline in actual employment. In addition, further diminished macrolevel profit and output dampens firms' ability to generate employment, which exacerbates the economic downturn. The long-lasting downward trend in private employment ($N_t^P$) begins to reverse around the 10th period, with a marked resurgence in the supply of corporate positions ($V_t^P$) and a concurrent rise in unemployment ($U_t^P$). In Scenarios 3–5, the public sector job hunting rate quickly returns to a steady state, preventing a prolonged decline in labor supply for companies. Additionally, firms lower wages flexibly, reducing the extent of job cuts. As a result, the decrease in new employment is significantly less than in the first two scenarios, and the employment level recovers to pre-shock levels in a relatively short time.

Fig 1(E) illustrates the response of the overall unemployment rate, which is the result of simultaneous changes in the new employment in both sectors ($N_t^i (i = P, G)$). In Scenario 1, firm employment falls and the public sector does not plan to expand, rendering many job seekers unable to find jobs and prolonging job searching status. In Scenario 2, workers' strong wage negotiation capabilities impede corporate wage reduction efforts, decreasing new private employment more than in Scenario 1. Although the public sector provides more job opportunities, the employment elasticity of the two sectors ($dln\, (V_{t+1}^G)/dln\, (V_t^P)$) being -1 indicates that the reduction in corporate jobs substantially exceeds the increased public sector jobs due to the larger base of the private sector. Consequently, the public sector can only absorb a fraction of the unemployed, driving up the overall unemployment rate. In Scenarios 3–5, firms' success in reducing labor cost prevents a continuous decline in the unemployment rate. In Scenario 5, corporations maintain employment levels through low-cost measures, and the public sector also contributes by actively expanding job offers, reaching the lowest unemployment rate.

In summary, the implementation of non-cyclical wage policies in Scenarios 1 and 2 severely weakens corporate employment, resulting in a long-term decline in overall employment. The increase in unemployment is characterized by a long-term duration and significant magnitude, and fails to revert to the preshock steady state value after the dissipation of the exogenous shock's impact. Instead, it shifts to a higher equilibrium level. Under these two rigid employment policies, the public sector contributes to the rise in long-term unemployment rather than having a positive influence as an economic stabilizer or cushion.

## 6 Conclusion and policy implication

The size and wages of public sector employment in China have been directly controlled by the government for a long time, making them less susceptible to market changes and economic fluctuations. Based on a directed search matching model and utilizing DSGE simulations, this study finds that the public sector's rigid employment policies have a negative effect on employment and economic stability, potentially acting as an amplifier of economic fluctuations rather than a stabilizer. The main reasons for this outcome are as follows. The stability of income in public sector positions raises (or at least stabilizes) job seekers' valuation of unemployment, enhancing their bargaining power with private firms. If the corporate sector cannot offer jobs with equivalent returns, workers will choose to remain unemployed and continue to seek positions in the public sector. Particularly when the economy faces negative external shocks, the public sector's rigid wage system will hinder firms from reducing costs to cope with shocks, suppressing their motivation for job creation. It also increases the number of potential job seekers that prefer public sector positions, encourages workers to extend voluntary unemployment, and exerts a significant crowding-out effect on corporate employment. Ultimately, this exacerbates the downward economic trend, amplifies economic cycle fluctuations, and may even lead to a rise in the natural unemployment rate. Conversely, if public sector wages are adjusted flexibly in response to economic cycles, whether maintaining the stability of public sector employment size or reducing it to alleviate government fiscal burdens, public sector employment can effectively function as an automatic stabilizer for employment and the macroeconomy.

In general, analyzing public sector employment in China from a purely economic perspective faces numerous challenges. The greatest difficulty, beyond the challenges of data acquisition and the insufficient depth of existing literature on search matching functions, is the traditional perspective that public sector employment lacks value for economic theoretical research under China's current centralized personnel management system as its political and social significance far outweighs its economic impact [138]. It is unrealistic to fully marketize government employment in China or use it as a fiscal policy tool to maintain employment and macroeconomic stability. In other words, economic criteria cannot be the basis for formulating and adjusting public sector employment policies in China [139]. Nevertheless, the issues examined in this study still have policy value. First, although this study focuses on public sector employment, its conclusions are equally applicable to low-marketized state-owned enterprises or mega corporations in monopolistic industries, which have market segmentation capabilities similar to the government and tend to isolate employment scale and remuneration from economic conditions [140]. Promoting the market-oriented reform of employment in such sectors and eliminating the negative impacts of rigid wages and employment is equally significant for macroeconomic stability. Additionally, this study provides a theoretical basis for adjusting civil servant wages. Due to the high public expectations of government welfare, reasonable salary increases in the public sector are often met with strong public discontent. Although this study does not propose a specific technical guideline for civil servant wage formulation, it

provides theoretical guidance for determining its adjustment direction following economic cycles. Simultaneous wage increases by the government and enterprises are reasonable during economic upturns or stable periods. This approach provides incentives and prevents corruption and creates the conditions for establishing a stable economic environment and expanding employment in alignment with the common interests of public and nonpublic sector workers.

It must be acknowledged that this study has certain limitations. First, there may be inaccuracies in assessing the public sector wage premium. Civil servant income in China has long been an elusive mystery to society. Civil servants often complain about low wages and demand raises, and compensation varies considerably across different levels and regions; however, the general public, particularly job seekers, are more aware of the numerous implicit benefits of civil service employment such as housing subsidies, medical and retirement security, childcare and schooling, and household registration solutions, in addition to potential sources of gray income from favors and misuse of public power. Considering these factors, it is difficult to objectively and accurately estimate the different wage premiums in public and private sectors [141]. Additionally, due to a lack of openness and transparency of Chinese employment data, important parameters in S&M models such as recruitment costs, turnover rates, and matching efficiency must be calibrated referencing foreign research. These issues might limit the accuracy of model predictions. Second, the study does not comprehensively consider other potential channels through which public sector employment impacts macroeconomic stability. For instance, the public sector might intervene in the private sector production, with substitution effects on private products, services, and investments, crowding out private sector employment [142]. The expansion of public sector employment could increase administrative expenditure, alter the structure of fiscal spending, increase corporate tax burdens, or cause a contraction in infrastructure investment, subsequently distorting private sector productivity and labor demand. In addition, positive impacts cannot be ruled out such as the government intervening to mitigate market shocks by controlling core economic resources or improving the investment environment through increased public goods supply, which could stimulate private investment and employment growth.

Effective management of public sector employment and compensation is crucial for governments as it broadly impacts the overall labor market stability and economic resilience to shocks. Based on the findings of this study, we propose three policy recommendations.

First, it is essential for policymakers to effectively reform civil servant wage policies, establishing a normalized compensation incentive adjustment mechanism that aligns with market conditions. This shift will entail a transition from an administratively driven, high-pressure supervision model to one that naturally incentivizes employees based on performance, introducing a stable mechanism for civil servants' wage growth. This change would strengthen the connection between public sector employment and broader economic conditions, sending accurate and timely signals to the labor market and empower workers to make informed and rational decisions regarding employment opportunities across various sectors. Singapore's approach, in which civil servant salaries are periodically adjusted based on a predefined ratio that considers inflation and market benchmarks, can provide a useful reference. This method allows for salary adjustments that align with economic conditions, accelerating increases in times of economic prosperity and future optimism, while slowing or pausing them during economic downturns. Such practices can not only align wages with economic realities, but will also enhance public acceptance of wage reforms. Additionally, the administrative identity management system in competitive state-owned enterprises requires transformation. Beyond maintaining a minimal number of administrative positions, the compensation frameworks for senior management roles should progressively align with market standards. This adjustment

will ensure that the public sector remains competitive and equitable, further harmonizing public employment practices with market-driven environments.

Second, the complexity of civil servant wage reform necessitates balanced consideration of multiple factors such as government finances, employment stability, equal opportunities, and talent motivation. It is crucial to implement a performance-based wage system, regularly evaluating civil servants on task completion and quality, and directly tying salaries to performance outcomes. To prevent subjectivity or undue influence, clear and publicly accessible assessment criteria are essential. To enhance transparency in formulating and revising civil servant wage standards, clearly communicating the purposes and implications of reforms to the public is crucial. Engaging diverse perspectives and fostering public participation can develop broad consensus and minimize resistance to changes. Given living cost and economic development disparities across China's regions, it is logical to introduce regional differentiation in civil servant wages to ensure equitable treatment across regions. Beyond salary, government focus should also include fostering career development opportunities for civil servants. Strengthening training programs and providing varied professional growth pathways can enable civil servants to feel accomplished and fulfilled in their positions. Modern human resource management techniques should be used to optimize government operations. This includes promoting government business outsourcing by establishing principal–agent relationships, enhancing talent dispatch mechanisms, and streamlining the workforce by reducing less effective civil servant positions. These measures will contribute to a more dynamic and efficient public sector.

Finally, the trend of university students fervently pursuing civil service exams reflects a typical reaction to an evolving job market. Over time, this civil service exam fever may inadvertently distort higher education's objectives as regular curricula and exam schedules increasingly accommodate civil service exam prep. Furthermore, if universally accepted, the allure of perceived benefits from civil service positions could undermine societal value systems. To counter these trends, a multifaceted approach is necessary. Promoting diverse career paths, particularly in innovation and entrepreneurship, will broaden professional horizons for youth and cultivate a family and societal atmosphere that values diverse employment options. Understanding the nature and duties of civil service jobs is crucial for young individuals, moving beyond the allure of stability, prestige, and benefits to a deeper comprehension of the responsibilities and commitments required can mitigate the rush toward civil service exams that is driven by peer influence. Policy measures can further support this shift. Encouraging entrepreneurship among university students can invigorate the job market. Building facilities such as startup spaces, incubators, accelerators, and industrial parks is essential for nurturing innovation and entrepreneurship. Additionally, providing venture capital and subsidies for commercial insurance premiums can reduce the initial hurdles for new enterprises. Furthermore, refining household registration and social security systems to minimize disparities in benefits across industries will enhance equity, encouraging the pursuit of careers outside traditional civil service paths. Such reforms can ensure balanced treatment and foster a healthier, more diverse employment landscape.

## Author Contributions

**Conceptualization:** Xiaodi Zhang.

**Data curation:** Xiaodi Zhang.

**Visualization:** Xiaodi Zhang.

**Writing – original draft:** Xiaodi Zhang.

**Writing – review & editing:** Xiaodi Zhang.

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
