## [Decision Letter · Decision Letter 0]

16 Jun 2024

PONE-D-24-14055

Public sector employment rigidity and macroeconomic fluctuations: A DSGE simulation for China

PLOS ONE

Dear Dr. Zhang,

Thank you for submitting your manuscript to PLOS ONE. After careful consideration, we feel that it has merit but does not fully meet PLOS ONE’s publication criteria as it currently stands. Therefore, we invite you to submit a revised version of the manuscript that addresses the points raised during the review process.

We look forward to receiving your revised manuscript.

Kind regards,

David Alaminos

Academic Editor

PLOS ONE

Journal Requirements:

   "The paper received funding for this work provided by National Natural Science Foundation of China."

Reviewers' comments:

Reviewer's Responses to Questions

**Comments to the Author**

1. Is the manuscript technically sound, and do the data support the conclusions?

Reviewer #1: Partly

Reviewer #2: Yes

2. Has the statistical analysis been performed appropriately and rigorously? 

Reviewer #1: Yes

Reviewer #2: Yes

3. Have the authors made all data underlying the findings in their manuscript fully available?

Reviewer #1: Yes

Reviewer #2: Yes

4. Is the manuscript presented in an intelligible fashion and written in standard English?

Reviewer #1: No

Reviewer #2: Yes

5. Review Comments to the Author

Reviewer #1: The topic selection of this paper is generally desirable and worthy of study. However, this paper argues that there is an inevitable correlation between China's weak GDP growth and public sector employment, which needs the support of multi-disciplinary research, and it is difficult to draw a conclusion through a set of data. For example, there are also large cultural factors that people expect public sector employment in China. The conclusions of this paper and policy recommendations should be discussed separately, and it is hoped to write specific articles, such as the first, second, third, etc. The use of language in this article also needs to be greatly revised, and it is recommended to hire professional organizations to assist. The format of this article is also different from the journal standard, please pay attention to the modification.

Reviewer #2: - The chosen topic for this paper is highly relevant and merits thorough investigation. The issue addressed is of significant importance, and the research has the potential to contribute valuable insights to the field. However, the assertion that there is an inevitable correlation between China's weak GDP growth and public sector employment requires robust support from multidisciplinary research. It is crucial to acknowledge that deriving a definitive conclusion from a single dataset is challenging and may not capture the complexity of the issue.

- It is essential to enrich the existing literature by incorporating insights from the following articles:

Alaminos, D. (2021). Factor Augmented Artificial Neural Network vs Deep Learning for Forecasting Global Liquidity Dynamics. In: Rutkowski, L., Scherer, R., Korytkowski, M., Pedrycz, W., Tadeusiewicz, R., Zurada, J.M. (eds) Artificial Intelligence and Soft Computing. ICAISC 2021. Lecture Notes in Computer Science, vol 12854. Springer, Cham.

Alaminos, D., Salas, M. B., & Fernández-Gámez, M. A. (2024). Global patterns and extreme events in sovereign risk premia: a fuzzy vs deep learning comparative. Technological and Economic Development of Economy, 30(3), 753–782.

Alaminos, D., Becerra-Vicario, R., Cisneros-Ruiz, A.J., Solano-Sánchez, M.Á. (2020). Estimating Optimal Military Spending Policy in DSGE Model: Empirical vs Theoretical Approach. Journal of Scientific & Industrial Research, 79(3), 193-196.

Alaminos, D., Salas, M.B. (2023). Tourism Stock Prices, Systemic Risk and Tourism Growth: A Kalman Filter with Prior Update DSGE-VAR Model. In: Rutkowski, L., Scherer, R., Korytkowski, M., Pedrycz, W., Tadeusiewicz, R., Zurada, J.M. (eds) Artificial Intelligence and Soft Computing. ICAISC 2022. Lecture Notes in Computer Science, vol 13589. Springer, Cham.

Alaminos, D., Ramírez, A., Fernández-Gámez, M.A., Becerra-Vicario, R. (2020). Estimating DSGE Models using Multilevel Sequential Monte Carlo in Approximate Bayesian Computation. Journal of Scientific & Industrial Research, 79(1), 21-25.

Alaminos, D., León-Gómez, A., Fernández-Gámez, M.A., Ferreira, T.S. (2020). Next Reaction Method for Solving Dynamic Macroeconomic Models: A Growth Regressions Simulation. Journal of Scientific & Industrial Research, 79(4), 277-280.

- The argument presented in this paper highlights the need for a multidisciplinary approach to fully understand the relationship between China's GDP growth and public sector employment. Factors such as cultural expectations towards public sector employment in China play a significant role and must be considered. Future research should incorporate perspectives from economics, sociology, and cultural studies to provide a more comprehensive analysis and to substantiate the findings with a broader evidence base.

- It is recommended that the conclusions and policy recommendations be discussed in separate sections to enhance clarity and focus. This approach allows for a more detailed exploration of each aspect and facilitates a structured presentation of the study's implications. Additionally, creating distinct articles for each major point, such as the first, second, and third recommendations, can provide a deeper and more nuanced discussion, making the research more accessible and actionable for policymakers and other stakeholders.

6. PLOS authors have the option to publish the peer review history of their article (what does this mean?). If published, this will include your full peer review and any attached files.

Reviewer #1: **Yes: **bangfan liu

Reviewer #2: No

---

## [Author Response · Author response to Decision Letter 0]

14 Jul 2024

Dear Editor and Reviewers, 

I express my profound gratitude to the esteemed editors and two diligent reviewers for their invaluable contributions in the form of constructive comments and suggestions. The opportunity to revise this paper is greatly appreciated. In response to the review team’s suggestions, I have meticulously addressed each comment and suggestion in a detailed and methodical manner, offering comprehensive responses in the section below. I hope this effort meets your satisfaction and that you will find the revised manuscript suitable for publication in PLOS ONE.

For ease of review, the main changes in the revised manuscript are marked in BLUE (Kind reminder: To keep the page clean, we chose not to use the 'Track Changes' feature in MS Office Word, and instead, set the text of the modifications to blue). I hope that these revisions are satisfactory. Thank you very much for your assistance with this paper.

Response to the Editor

Editor point 1: Please ensure that your manuscript meets PLOS ONE's style requirements,including those for file naming.

The PLOS ONE style templates can be found at

https://journals.plos.org/plosone/s/fle?id=ba62/PLOSOne_formatting_sample_title_authors_affiliations.pdf

Response: I sincerely appreciate your comments and the opportunity to revise my manuscript. I have carefully formatted the manuscript to fully comply with the PLOS ONE style requirements. In the revised manuscript, I have used double spacing as per the submission guidelines. I hope this does not hinder your reading due to the wider line spacing.

Editor point 2: Please note that PLOS ONE has specific guidelines on code sharing for submissions in which author-generated code underpins the findings in the manuscript.In these cases,all author-generated code must be made available without restrictions upon publication of the work.Please review our guidelines at

https://journals.plos.org/plosone/s/materials-and-software-sharing#loc-sharing-code and ensure that your code is shared in a way that follows best practice and facilitates reproducibility and reuse.

Response: Thank you for highlighting the importance of code sharing for reproducibility and adherence to PLOS ONE guidelines. I have uploaded the code to a publicly accessible Figshare repository, which does not impose any access restrictions: https://doi.org/10.6084/m9.figshare.26299456. I’m also more than happy to provide the link to the repository in the manuscript, ensuring that review team and readers can access and use the code without restrictions upon the publication of our work. I believe that this fulfills the best practices for code sharing as recommended by PLOS ONE.

Editor point 3: Please note that funding information should not appear in any section or other areas of your manuscript. We will only publish funding information present in the Funding Statement section of the online submission form.Please remove any funding-related text from the manuscript.

Response: I have removed all funding-related text from the manuscript as per your instructions. The details regarding funding have been included exclusively in the Funding Statement section of the online submission form, in compliance with PLOS ONE’s guidelines.

Editor point 4: We note that the grant information you provided in the`Funding Information' and `Financial Disclosure sections do not match. When you resubmit,please ensure that you provide the correct grant numbers for the awards you received for your study in the `Funding Information'section.

Response: Thank you for pointing out the discrepancies in the grant information. I have reviewed and corrected the grant numbers to ensure consistency. All relevant and accurate grant details for this study have been updated.

Editor point 5: Thank you for stating the following financial disclosure:

"The paper received funding for this work provided by National Natural Science Foundation of China."

Please state what role the funders took in the study. If the funders had no role, please state:"The funders had no role in study design, data collection and analysis,decision to publish,or preparation of the manuscript." If this statement is not correct you must amend it as needed.

Response: In response to your request, I confirm that “The funders had no role in study design, data collection and analysis, decision to publish, or preparation of the manuscript.” I will include this amended Role of Funder statement in the cover letter to ensure consistency across all documentation.

I extend my heartfelt gratitude for your dedicated and diligent efforts in evaluating my manuscript. I earnestly hope that the revisions I have made will meet your expectations and receive your approval. 

Response to Reviewer 1

General comment: The topic selection of this paper is generally desirable and worthy of study.

Response: I really appreciate your interest and helpful comments on my research. This article is dedicated to providing a comprehensive analysis of how public sector employment interact with macroeconomic dynamics in China, and providing theoretical support for establishing a flexible wage adjustment mechanism linked to market conditions in the public sector. I believe this research motivation is meaningful and necessary.

Comment 1: This paper argues that there is an inevitable correlation between China's weak GDP growth and public sector employment, which needs the support of multi-disciplinary research, and it is difficult to draw a conclusion through a set of data. For example, there are also large cultural factors that people expect public sector employment in China. 

Response: I fully agree with your point that what Chinese people expect from the public sector employment needs the support of multi-disciplinary research. It is also not rigorous to assert a correlation between public sector employment in China and the economic slowdown after the pandemic based solely on a single set of data. I really appreciate the opportunity to provide further clarification on this important point. In response to your valuable critique, I have made the following revisions to the manuscript：

Multi-Disciplinary Background Analysis: As you pointed out, understanding the unique preference for public sector employment among Chinese job seekers requires a thorough analysis from historical, cultural, economic, political, and sociological perspectives. To this end, I have expanded the literature review sections (L48-126 of Revised Manuscript with Track Changes) to systematically outline these factors. This approach aims to enrich the contextual background of our study without diverting from the main economic focus of the paper.

Avoiding Deterministic Statements: I have carefully revised the manuscript to eliminate any assertive statements that may suggest a direct correlation or causation between public sector employment and GDP growth without empirical evidence (specifically, rewrite the two sentences in L4-7, L26-27 of Revised Manuscript with Track Changes). Instead, the revised text highlights the observable increase in public sector job seekers post-COVID-19 as an introductory context to the broader discussion of employment rigidity and its macroeconomic implications. These adjustments ensure that our conclusions remain strictly within the scope of the theoretical framework and simulation discussed.

Comment 2: The conclusions of this paper and policy recommendations should be discussed separately, and it is hoped to write specific articles, such as the first, second, third, etc. 

Response: Many thanks for bringing this to my attention. I have rewritten the “Conclusion and policy implication” section of the paper to discuss the conclusions and policy recommendations in separate sections. Each policy recommendation is now clearly numbered as first, second, third, etc., to facilitate clear delineate the implications and actionable steps derived from the research findings (L718-772 of Revised Manuscript with Track Changes).

Comment 3: The use of language in this article also needs to be greatly revised, and it is recommended to hire professional organizations to assist. 

Response: Thanks for kindly reminding me of this point. I’m also aware that the English writing of the paper was not satisfactory. Therefore, I have hired the Elsevier Language Editing services (Order reference ASLESTD1066891). I chose this organization primarily for its authoritative reputation in the field and its affordability. The Certificate of Elseviere has been uploaded in “Other” materials. Furthermore, I have diligently proofread the manuscript, paying particular attention to rectifying any grammatical errors and enhancing the overall quality of the language used. I hope the revised manuscript aligns with the stringent standards set by PLOS ONE.

Comment 4: The format of this article is also different from the journal standard, please pay attention to the modification.

Response: Thank you for drawing attention to the formatting issue. I have carefully reviewed the journal’s guidelines and have made all necessary adjustments to the format of the manuscript. This includes updating the structure, headings, citations, and any other elements to align with the required standards.

Again, I highly appreciate the opportunity to revise and resubmit the manuscript again as well as your comments and suggestions on it. They have helped to improve the manuscript significantly. I hope I have answered all of your questions and have revised the manuscript to your satisfaction.

Response to Reviewer 2

General comment: The chosen topic for this paper is highly relevant and merits thorough investigation. The issue addressed is of significant importance, and the research has the potential to contribute valuable insights to the field. 

Response: I greatly appreciate your recognition of the relevance and importance of the chosen topic. I am committed to thoroughly addressing all of your comments and suggestions to enhance the quality of this manuscript. 

Comment 1: However, the assertion that there is an inevitable correlation between China's weak GDP growth and public sector employment requires robust support from multidisciplinary research. It is crucial to acknowledge that deriving a definitive conclusion from a single dataset is challenging and may not capture the complexity of the issue.

Response: Many thanks for your insightful advice. I agree that it is inappropriate to assert a definitive causal relationship between China's economic downturn and the increasing preference for public sector employment without empirical evidence. Consequently, I have made revisions to the manuscript to avoid such conclusive viewpoints. Some assertive statements that may suggest a correlation (negative or otherwise) between public sector employment and economic growth have been rewritten (two sentences in L4-7, L26-27 of Revised Manuscript with Track Changes). 

Since the pandemic, there has indeed been a noticeable increase in the number of people in China taking or intending to take civil service exams; some data has been added to the manuscript as supporting information (L09-120 of Revised Manuscript with Track Changes). However, this paper does not aim to discuss a strong causality between the two, but rather uses this "phenomenon" as a background to explore how the rigidity of public sector employment affects macroeconomic cycles.

Comment 2: It is essential to enrich the existing literature by incorporating insights from the following articles:

Alaminos, D. (2021). Factor Augmented Artificial Neural Network vs Deep Learning for Forecasting Global Liquidity Dynamics. In: Rutkowski, L., Scherer, R., Korytkowski, M., Pedrycz, W., Tadeusiewicz, R., Zurada, J.M. (eds) Artificial Intelligence and Soft Computing. ICAISC 2021. Lecture Notes in Computer Science, vol 12854. Springer, Cham.

Alaminos, D., Salas, M. B., & Fernández-Gámez, M. A. (2024). Global patterns and extreme events in sovereign risk premia: a fuzzy vs deep learning comparative. Technological and Economic Development of Economy, 30(3), 753–782.

Alaminos, D., Becerra-Vicario, R., Cisneros-Ruiz, A.J., Solano-Sánchez, M.Á. (2020). Estimating Optimal Military Spending Policy in DSGE Model: Empirical vs Theoretical Approach. Journal of Scientific & Industrial Research, 79(3), 193-196.

Alaminos, D., Salas, M.B. (2023). Tourism Stock Prices, Systemic Risk and Tourism Growth: A Kalman Filter with Prior Update DSGE-VAR Model. In: Rutkowski, L., Scherer, R., Korytkowski, M., Pedrycz, W., Tadeusiewicz, R., Zurada, J.M. (eds) Artificial Intelligence and Soft Computing. ICAISC 2022. Lecture Notes in Computer Science, vol 13589. Springer, Cham.

Alaminos, D., Ramírez, A., Fernández-Gámez, M.A., Becerra-Vicario, R. (2020). Estimating DSGE Models using Multilevel Sequential Monte Carlo in Approximate Bayesian Computation. Journal of Scientific & Industrial Research, 79(1), 21-25.

Alaminos, D., León-Gómez, A., Fernández-Gámez, M.A., Ferreira, T.S. (2020). Next Reaction Method for Solving Dynamic Macroeconomic Models: A Growth Regressions Simulation. Journal of Scientific & Industrial Research, 79(4), 277-280.

Response: I am deeply grateful for the literature references you provided for this research. I have carefully reviewed these articles and cited them in proper parts of the new manuscript (L177-191, Reference 93-08 of Revised Manuscript with Track Changes). I believe they are helpful in improving the quality of the literature review part. 

The inclusion of the 6 articles not only introduces readers to the latest technical advancements in DSGE modeling but also demonstrates the substantial contributions of combining DSGE models with advanced estimation techniques and algorithms in macroeconomic analysis. This significantly supports the rationale for applying DSGE methods in my research. Furthermore, cutting-edge techniques from the literature, such as the Next Reaction Method, Multilevel Sequential Monte Carlo, and Deep Learning, have greatly benefited my own pursuit of studying marcro models. I have gained a deeper understanding of the state of the art trends and emerging hotspots in this field. The fresh insights and methodologies presented in these articles have inspired new avenues of inquiry that I might not have previously considered.

The article Estimating Optimal Military Spending Policy in DSGE Model: Empirical vs Theoretical Approach provides a robust methodological approach by comparing DSGE, VAR, and DSGE-VAR models. It helps to demonstrate the benefits of combining theoretical and empirical approaches to achieve more accurate estimates. 

The article Next Reaction Method for Solving Dynamic Macroeconomic Models: A Growth Regressions Simulation introduces a novel computational technique for improving the precision and reliability of macroeconomic model estimates, particularly in handling measurement errors and residual correlations.

The article Estimating DSGE Models using Multilevel Sequential Monte Carlo in Approximate Bayesian Computation provides techniques that is useful in achieving higher accuracy levels and reduced computational costs, especially in dealing with small and irregular data samples.

The article Tourism Stock Prices, Systemic Risk and Tourism Growth: A Kalman Filter with Prior Update DSGE-VAR Model demonstrates the application of a Bayesian Kalman Filter with Prior Update (BKPU) to increase the robustness of DSGE and VAR models. The techniques discussed were applied to enhance the stability and accuracy of the economic models, particularly in understanding the financial variables.

The article Factor Augmented Artificial Neural Network vs Deep Learning for Forecasting Global Liquidity Dynamics demonstrates the effectiveness of combining traditional econometric methods with machine learning techniques, which can be applied to enhance the prediction models in my research on public sector employment dynamics. 

The article Global patterns and extreme events in sovereign risk premia: a fuzzy vs deep learning comparative

---

## [Editor Report · Decision Letter 1]

26 Jul 2024

Public sector employment rigidity and macroeconomic fluctuation: A DSGE simulation for China

PONE-D-24-14055R1

Dear Dr. Zhang,

We’re pleased to inform you that your manuscript has been judged scientifically suitable for publication and will be formally accepted for publication once it meets all outstanding technical requirements.

Kind regards,

David Alaminos

Academic Editor

PLOS ONE
---

## [Editor Report · Acceptance letter]

30 Jul 2024

PONE-D-24-14055R1 

PLOS ONE

Dear Dr. Zhang, 

I'm pleased to inform you that your manuscript has been deemed suitable for publication in PLOS ONE. Congratulations! Your manuscript is now being handed over to our production team.

Kind regards, 

on behalf of

Dr. David Alaminos 

Academic Editor

PLOS ONE